# Utilizing the Metaverse to Provide Innovative Psychosocial Support for Pediatric, Adolescent, and Young Adult Patients with Rare Cancer

**DOI:** 10.3390/cancers16152617

**Published:** 2024-07-23

**Authors:** Joe Hasei, Hisashi Ishida, Hideki Katayama, Naoko Maeda, Akihito Nagano, Motoharu Ochi, Masako Okamura, Shintaro Iwata, Kunihiro Ikuta, Shinichirou Yoshida, Tomohiro Fujiwara, Eiji Nakata, Ryuichi Nakahara, Toshiyuki Kunisada, Toshifumi Ozaki

**Affiliations:** 1Department of Medical Information and Assistive Technology Development, Graduate School of Medicine, Dentistry and Pharmaceutical Sciences, Okayama University, Okayama 700-0082, Japan; 2Department of Pediatrics, Okayama University Hospital, Okayama 700-8558, Japan; 3Department of Palliative and Supportive Care, Okayama University Hospital, Okayama 700-8558, Japan; 4Department of Pediatrics, NHO National Hospital Organization Nagoya Medical Center, Nagoya 460-0001, Japan; 5Department of Orthopedic Surgery, Graduate School of Medicine, Gifu University, Gifu 501-3822, Japan; 6Division of Survivorship, Institute for Cancer Control, National Cancer Center, Tokyo 104-0045, Japan; 7Department of Musculoskeletal Oncology and Rehabilitation, National Cancer Center Hospital, Tokyo 104-0045, Japan; 8Department of Orthopedic Surgery, Graduate School of Medicine, Nagoya University, Nagoya 464-0083, Japan; 9Department of Orthopedic Surgery, Graduate School of Medicine, Tohoku University, Sendai 980-0872, Japan; 10Science of Functional Recovery and Reconstruction, Graduate School of Medicine, Dentistry and Pharmaceutical Sciences, Okayama University, Okayama 700-0082, Japan

**Keywords:** virtual reality, metaverse, adolescent and young adult, rare cancer, mental health

## Abstract

**Simple Summary:**

Pediatric, adolescent, and young adult (AYA) cancer patients, especially those with rare cancers, face unique challenges that traditional support systems may not adequately address. This study explores the potential of the metaverse, a virtual shared space, to provide psychological support and foster a sense of community among these patients. By utilizing customizable avatars and facilitating interactions across geographical and temporal barriers, the metaverse enables patients to connect with peers, share experiences, and receive emotional support. The anonymity provided by avatars helps reduce appearance-related anxiety and the stigma associated with cancer treatment. This research highlights the metaverse’s potential to transform the delivery of psychosocial support for patients with rare cancers, offering a novel approach to patient care that transcends physical limitations and fosters a global patient community. The findings suggest that integrating virtual spaces into healthcare models can effectively meet the complex needs of pediatric and AYA cancer patients.

**Abstract:**

This study investigated the potential of the metaverse in providing psychological support for pediatric and AYA cancer patients, with a focus on those with rare cancers. The research involved ten cancer patients and survivors from four distinct regions in Japan, who participated in metaverse sessions using customizable avatars, facilitating interactions across geographical and temporal barriers. Surveys and qualitative feedback were collected to assess the psychosocial impact of the intervention. The results demonstrated that the metaverse enabled patients to connect with peers, share experiences, and receive emotional support. The anonymity provided by avatars helped reduce appearance-related anxiety and stigma associated with cancer treatment. A case study of a 19-year-old male with spinal Ewing’s sarcoma highlighted the profound emotional relief fostered by metaverse interactions. The findings suggest that integrating virtual spaces into healthcare models can effectively address the unique needs of pediatric and AYA cancer patients, offering a transformative approach to delivering psychosocial support and fostering a global patient community. This innovative intervention has the potential to revolutionize patient care in the digital age.

## 1. Introduction

Pediatric, adolescent, and young adult (AYA) patients with cancer face multifaceted challenges that extend beyond the physical toll of the disease and encompass psychological and social hurdles [1]. The interruption of education or work and altered relationships with friends and family underscore the complexity of their predicaments. Moreover, rare cancers are prevalent among pediatric and AYA patients [2], limiting their opportunities to interact with their peers and leading to feelings of isolation and alienation [3]. In the United States, nearly 89,000 AYA patients are diagnosed with cancer annually, and many report feeling lonely [4]. AYA patients desire to hear personal and subjective experiences to explore solutions to potential future challenges [5]. Peer support among patients with the same condition can significantly empower patients with cancer by providing information and psychological support [6,7]. Despite the clear need for peer support among pediatric and AYA patients with cancer, several challenges arise in providing effective support programs.

Peer support for cancer patients under the same conditions can provide invaluable information and psychological support. However, several reviews examining the effectiveness of peer support programs have emphasized the need to address psychological outcomes concerning socioeconomic factors, such as marital status, education, and income level [8,9]. These findings suggest that the impact of peer support may vary depending on the participants’ socioeconomic backgrounds. Constructing peer support groups with various patient backgrounds is challenging for rare diseases. This is especially true for the pediatric and AYA generations, as even minor age differences lead to completely different social backgrounds, such as between elementary school, middle school, high school, university, and working life. Consequently, adolescents and young adults with cancer and their parent caregivers have high levels of unmet healthcare service needs that are associated with more significant emotional distress [10]. The diverse social contexts within this age group make it challenging to provide adequate support to address their unique concerns and experiences. This highlights the importance of tailored interventions to bridge these gaps and meet their needs. Therefore, a system that recruits participants nationwide is necessary to facilitate interactions among patients with similar conditions.

Peer support, a vital source of empowerment that provides invaluable information and psychological solace to AYA patients, is crucial because unmet service needs in this population are associated with worse health-related quality of life [11]. The variability in patient backgrounds, especially among rare diseases, makes forming inclusive support groups daunting. These considerations underscore the need for a novel approach that transcends geographical and demographic constraints to facilitate meaningful connections between patients with shared experiences.

The metaverse is an innovative realm that leverages avatar-based interactions to foster community and support among pediatric patients with AYA cancer. This study explores the potential of metaverse-mediated interactions to alleviate appearance-related anxiety, enhance emotional expression, and improve communication with healthcare providers using avatars. Generally, avatars play a crucial role in the metaverse, allowing patients to create digital representations that help them feel more comfortable expressing their emotions and connecting with others [12], significantly increasing social presence and interpersonal trust, facilitating open communication, and fostering community within a virtual environment [13]. Furthermore, the anonymity provided by avatars can help to reduce the stigma and self-consciousness often associated with rare cancers, enabling patients to connect more easily with their peers. Avatar-based interactions have been shown to yield multifaceted effects, including reduced appearance-related anxiety, enhanced emotional expression, and improved communication with healthcare providers [14]. Therefore, the metaverse offers a unique platform for patients with rare cancers to share treatment-related information and experiences, thereby providing invaluable psychological support.

The introduction of metaverse-mediated interactions represents a groundbreaking approach to patient support, marking the first endeavor to harness this technology to enhance the well-being of patients with cancer. This study aims to demonstrate the innovative potential of metaverse technology in providing psychological support and fostering a sense of community among pediatric and AYA patients with cancer, particularly those with rare cancers. By establishing a foundation for future research, this study highlights the potential of metaverse spaces to transform patient care and provide support and optimism for those facing the challenges of cancer treatment and recovery.

## 2. Materials and Methods

### 2.1. Participants

This study involved ten cancer patients and survivors, aged 10–22 years, from four distinct oncology centers across Japan, representing both urban and rural areas, including isolated areas. This approach ensured comprehensive demographic representation and captured a broad spectrum of patient experiences. The inclusion criteria were as follows: (1) confirmed diagnosis of cancer and (2) willingness to participate in virtual interactions. The exclusion criteria included (1) acute medical conditions precluding participation and (2) lack of parental consent for minors. Primary oncologists at each center identified potential participants based on these criteria among pediatric, adolescent, and young adult patients with cancer. The research team subsequently approached eligible patients and their families for enrollment.

### 2.2. Procedure (Settings and Recruitment)

Initial contact with potential participants and their families was established by the treating oncologists at each participating hospital. The research team then conducted sessions at hospitals to explain the study’s objectives and procedures to interested families. These initial briefings were held to inform patients and their guardians about the objectives and framework of the metaverse interaction sessions. Following these informative sessions, informed consent was obtained from all participants. Detailed demographic and clinical data, such as age, sex, specific diagnosis, and initial cancer site, were meticulously gathered and securely entered into a cloud-based database. Participant selection was contingent upon their continued willingness to engage after potential matches were identified based on demographic and clinical data. Scheduled sessions were conducted only after securing consent, adhering to stringent ethical guidelines approved by the institutional review boards of all participating facilities.

### 2.3. Intervention

The metaverse space used in this study was specifically designed for this research through the collaborative effort of medical professionals and metaverse designers. The virtual environment was developed using Unity as a world within “cluster”, a metaverse platform provided by Cluster, Inc. (Shinagawa, Tokyo, Japan) This platform was chosen for its accessibility across various devices, including smartphones, PCs, and VR headsets, ensuring that all participants could engage regardless of their physical limitations or preferred mode of access. The initial 3D model of the space was created by professional metaverse designers. This basic framework was refined and customized by a multidisciplinary medical team consisting of specialists in bone and soft tissue tumors, pediatric oncology, and palliative care. These medical experts were responsible for the detailed arrangement of elements within the space and the creation of specific content tailored to the needs of pediatric and AYA cancer patients. The resulting virtual environment was enriched with natural elements, aiming to mitigate the lack of natural exposure for patients predominantly confined to hospital settings by offering a virtual yet immersive natural experience (Figure 1a). The platform’s features, including real-time voice chat, emoticons, and gesture animations, were utilized to enhance communication and interaction among participants. The method of accessing the metaverse varied depending on each participant’s circumstances. For inpatients, those in private rooms participated from their hospital rooms, whereas those in shared rooms were moved to separate private spaces for the sessions. All inpatients used VR goggles provided by the research team to access the metaverse device. These devices were equipped with the “cluster” metaverse platform application provided by Cluster, Inc. The application was consistently updated to the latest available version throughout the study period to ensure optimal performance and security. The devices were configured to ensure ease of use and privacy for the participants. Survivors who were no longer hospitalized participated from their homes using their own smartphones. The sessions began with icebreaking activities that involved interacting with various elements of the virtual space, designed to alleviate initial anxiety and foster a comfortable atmosphere for open communication. Physicians and psychologists, serving as facilitators, ensured that discussions remained supportive and constructive throughout. To maintain a non-intimidating environment, staff were uniformly represented as small dogs (Figure 1b). Each session lasted 40 min, allowing free-form conversation among the participants.

### 2.4. Measures

Following each session, a survey was conducted to collect evaluations and feedback from the participants. The survey consisted of items rated on a five-point Likert scale and included open-ended questions to allow for qualitative feedback. This data collection aimed to measure the psychosocial impacts of the intervention.

### 2.5. Analysis

The data collected from the surveys were analyzed to assess the impact of metaverse-mediated interactions on the psychosocial well-being of patients. The analysis primarily focused on how the intervention influenced patient engagement and addressed their unmet healthcare needs. Additionally, the study explored the potential therapeutic benefits of participation in a naturalistic virtual environment. Both quantitative data from survey responses and qualitative feedback from open-ended questions were thoroughly evaluated. The qualitative assessment provided deep insights into participants’ emotions and experiences, allowing for a comprehensive understanding of how the use of the metaverse impacted their psychosocial experiences.

## 3. Results

Based on the recruitment criteria outlined in the Materials and Methods section, the following participants were enrolled in this study. The study enrolled ten cancer patients and survivors from four distinct regions in Japan, aiming to capture a broad spectrum of experiences across both urban and rural settings. Table 1 summarizes the participant characteristics, including age, diagnosis, treatment status, and city of residence, categorized into three major geographical regions of Japan: Central, East, and West. These regional classifications are used to facilitate the understanding of diverse patient experiences influenced by geographical factors.

The regions specified in this table represent broad geographical areas of Japan, categorized as Central, East, and West Japan. This categorization aims to provide insights into the regional distribution of participants and is not intended for precise geographical analysis.

### 3.1. Initial Interactions in the Metaverse

During their initial encounters with the metaverse, all of the participants reported positive experiences and expressed enthusiasm for future participation. The participants naturally engaged in conversations about their shared experiences and concerns about their daily hospital lives. The discussions covered topics such as the assortment of items available at the hospital store, experiences during short-term home stays between treatment cycles, activities anticipated during such stays, and specific foods they craved outside the hospital. Additionally, the student participants reported that they communicated about their hospitalizations to friends and shared their thoughts on potential school trip destinations. These conversations laid the groundwork for building connections among the participants, providing an opportunity to exchange personal insights and advice. Discussing common concerns within the relaxed, game-like environment of the metaverse helped participants normalize their experiences and mitigate the feelings of isolation often associated with long-term hospitalization.

Moreover, the metaverse platform enabled participants to engage in a manner that closely resembled face-to-face interactions, facilitating the development of genuine relationships and a sense of community. This is particularly valuable for pediatric and AYA patients with cancer, who often face significant challenges in maintaining social connections owing to the nature of their treatment and the rarity of their conditions. These initial interactions in the metaverse highlight the platform’s potential to create a supportive and engaging environment for pediatric and AYA cancer patients. By fostering a sense of belonging and sharing experiences, the metaverse can significantly alleviate the psychological burden of cancer treatment and enhance patients’ overall well-being.

### 3.2. Case Study: Impact on Individual Patients

An impressive case involved a 19-year-old man diagnosed with thoracic spinal Ewing’s sarcoma. This rare cancer primarily affects pediatric and adolescent patients, with an annual incidence rate of only 1.59 cases per million [15]. This type of cancer is even less common in African, East Asian, and Southeast Asian populations [15]. Spinal manifestations are sporadic, accounting for only 8% of the cases [16], which significantly restricts the opportunities for patients to meet peers with similar diagnoses. Upon admission, this patient was administered an intensive chemotherapy regimen of Vincristine, Doxorubicin, Cyclophosphamide, and Ifosfamide Etoposide. Following admission, he experienced severe emotional distress marked by intense anxiety and deep loneliness, and he did not smile once during this challenging period. During treatment, a serendipitous connection occurred when a 20-year-old patient with lumbar spine Ewing’s sarcoma, living approximately 350 km away, joined the same metaverse platform. Despite their geographical distance, their similar ages, diagnoses, sites of onset, and sexes facilitated instant bonds. Within this digital realm, they shared their experiences and supported each other. The lumbar spine patient, who had just completed his scheduled course of chemotherapy, was provided practical advice on coping with the side effects. He discussed how the residual scent of medications often caused discomfort and found that chewing gum helped mitigate this unpleasant smell. He also recommended cold drinks as a solution for appetite loss. This invaluable exchange of coping strategies provided physical relief and essential emotional support. During these heartfelt exchanges, the thoracic spine patient smiled for the first time since hospitalization, illustrating the profound emotional relief that such interactions can foster. This encounter highlights the transformative potential of the metaverse as a platform for emotional and social support, offering a space in which patients can connect, share their hardships, and find solace among those who truly understand their struggles.

### 3.3. Survey Results: Ewing’s Sarcoma Patients’ Feedback

Three patients with Ewing’s sarcoma participated in this survey. Our decision to focus on these three adolescent patients diagnosed with Ewing’s sarcoma in this survey was driven by the extremely low incidence of this disease and the unique opportunity presented by their identical ages and genders. This rare alignment makes their insights particularly valuable for evaluating the impacts of our study. The surveys indicated that using avatars facilitated effortless conversations and improved emotional expressions, suggesting that interactions in the metaverse could serve as psychological support (Table 2).

Questionnaire responses from three patients with Ewing’s sarcoma. Each question was answered on a Likert scale (1. Strongly disagree, 2. Disagree, 3. Neither agree nor disagree, 4. Agree, 5. Strongly agree).

### 3.4. Benefits of Avatar-Mediated Communication in the Metaverse

This section summarizes the qualitative analysis of feedback received from participants regarding the benefits of avatar-mediated communication in the metaverse. The unique virtual setting of the metaverse allows for enhanced interaction and communication experiences, which have been explored through detailed feedback from patients. These insights have been instrumental in understanding how digital interactions can mitigate some of the common challenges faced by patients, particularly in terms of psychological comfort and social connectivity. The findings are based on a systematic collection and analysis of participants’ responses, highlighting the significant role of virtual environments in improving patient care and support. Avatar-mediated communication within the metaverse has been proven to significantly reduce stress and enhance comfort during interactions with healthcare professionals, supporting existing research on the utility of the metaverse in health communication [17]. Patient feedback underscores several crucial benefits of using the metaverse.

#### 3.4.1. Reduced Appearance-Related Anxiety

Patients experienced less anxiety about physical changes, such as hair loss and edema, which are common side effects of cancer treatments, due to anonymity and control over their virtual appearance.

#### 3.4.2. Improved Communication with Healthcare Providers

Using avatars and virtual environments facilitated more open discussions, making it easier for patients to discuss sensitive health issues without the discomfort of their physical presence.

#### 3.4.3. Opportunities for Natural Experiences and Social Interaction

The metaverse offers simulated environments, such as seas, sandy beaches, and lush greenery, enhancing the user’s experience and promoting social interactions in a more inviting setting.

#### 3.4.4. Psychological Stability through Empathy

Sharing experiences across the metaverse and receiving empathetic responses helped to stabilize the patients’ emotions and improve their overall psychological well-being.

The calming effect of the metaverse’s virtual natural settings contributed significantly to creating a conducive environment for stable and focused conversations. Patients reported that engaging in discussions in these soothing environments allowed them to converse with greater psychological comfort, enhancing the overall quality of communication. This innovative use of the metaverse presents a promising new platform for alleviating social isolation and improving peer support, particularly among pediatric and AYA patients with cancer.

## 4. Discussion

### 4.1. Differences between Metaverse Interaction and Traditional Communication Methods

Unlike traditional video calls where only one person can effectively speak at a time while others passively listen, metaverse interactions allow for dynamic subgroup conversations, mimicking real-life social gatherings. In the metaverse, participants can engage in multiple concurrent conversations with volume adjustments based on virtual proximity, significantly enhancing the natural flow and efficiency of group interactions (Figure 2a). Furthermore, the metaverse environment supports nonverbal communication through shared virtual experiences, such as collectively appreciating digital landscapes, interacting with virtual objects, and participating in group activities, such as dancing. This capacity for multimodal communication, which extends beyond spoken language, offers a liberating degree of interaction (Figure 2b). Moreover, the immersive nature of virtual reality, notably when employing VR headsets, has been reported to offer therapeutic benefits through exposure to natural virtual settings, thereby contributing positively to patient communication and stress alleviation. These features, which are difficult to achieve using online video calls, are unique to metaverse spaces.

### 4.2. Peer Support in the Metaverse

Peer support has been suggested to positively impact psychological adaptation to cancer diagnosis and treatment by directly reducing feelings of isolation, encouraging healthy behaviors, and promoting positive psychological states [18]. Both online and face-to-face peer support are adequate resources. However, challenges such as limited interaction among participants have been pointed out in group telephone-based peer support [19]. A significant advantage of the metaverse is its ability to allow conversations among subgroups within the same space, similar to real-world interactions.

### 4.3. Impact on the Well-Being of Patient Families

In the realm of rare pediatric diseases, parents often experience considerable physical and psychological stress as they seek disease-related information, and may become isolated in the process [20]. Peer support is a critical resource for informational and psychological support, with interactions between children in similar circumstances or between their parents being highly valued [21]. In contrast to traditional online forums and social media platforms, the metaverse offers an immersive and interactive environment. This setting facilitated real-time emotional connections with other families experiencing similar challenges, enabling deeper empathy and support. Consequently, this metaverse initiative has the potential to benefit not only patients but also their families. This system allows interactions among survivors, current inpatients, and their families, effectively meeting the needs of patients’ families for essential information about social life.

### 4.4. Virtual versus Real-World Peer Support: The Role of Avatars

Survivors who participated in this initiative expressed that conversing through avatars felt as natural as face-to-face communication, with no sense of incongruity, despite the anonymity provided by the metaverse. Using personally customized avatars enhances immersion and motivation in a virtual environment, suggesting that experiences in virtual settings can influence cognition and behavior in the real world [12]. Various forms of peer support, such as through face-to-face communication, telephone communication, and the Internet, have been described, but their efficacy varies, and there is no consensus [9]. The metaverse does not fall into these categories, representing a novel format that can be described as a fusion of online and face-to-face interactions. This innovative approach can potentially simulate real-world interactions within a virtual space, providing a unique blend that leverages the strengths of both realms. The psychosocial dimension of cancer, particularly stigma, presents a complex challenge that evolves throughout the stages of diagnosis and treatment [22], significantly affecting the quality of life and healthcare outcomes [23]. Furthermore, survivors who experience cancer-related stigma are more likely to leave their jobs than those who do not [24], making stigma management crucial for pediatric and AYA patients who face a long post-treatment period. Peer support, an integral component of the cancer care continuum, has demonstrated efficacy in reducing stigma, even when facilitated online [25]. Online-forum-based peer support allows participants to share their experiences openly without revealing their real-world identities, thus minimizing the perception of stigma [26]. However, while the anonymity inherent in online interactions brings substantial benefits, including a reduction in stigma and an increase in open communication, integrating face-to-face interactions is desirable. In-person engagements are unparalleled in their ability to convey empathy and offer quality advice directly, thus complementing the strengths of online platforms [26]. Consequently, by offering a unique blend of online and face-to-face peer support, the metaverse appears to be highly promising as a next-generation format for delivering more effective and impactful peer support interventions to cancer patients and survivors. This unique blend facilitates a safe space for open dialogue and promises enhanced patient engagement and outcomes through a more personalized and empathetic approach. One study involving higher-education students, a demographic similar in age to pediatric and AYA populations, found no difference in effectiveness between online and face-to-face peer support [27]. This finding suggests that the metaverse, with its unique blend of virtual and in-person interactions, is likely to be highly compatible with and appealing to younger cancer patients and survivors. The familiarity and comfort that younger generations have with digital platforms and virtual environments further underscore the potential of the metaverse to serve as an engaging and effective tool for delivering peer support interventions to this population. By leveraging the metaverse’s ability to create immersive and interactive experiences that resonate with the preferences and behaviors of pediatric and AYA patients, healthcare providers can foster greater patient engagement, improve psychosocial outcomes, and ultimately enhance the quality of life of young cancer survivors.

### 4.5. Alignment with Recent Innovative Approaches in Pediatric Care

Our findings align with recent research on innovative approaches to pediatric care. One report has demonstrated the efficacy of interactional mediation in enhancing medical consultations for children [28]. This study showed that children who engaged in interactive activities before medical consultations exhibited better cooperation during clinical care. This highlights the efficacy of the metaverse as a mediating platform for peer support, as both approaches aim to facilitate effective communication and cooperation in pediatric healthcare settings. The metaverse extends this concept by providing a virtual environment that can overcome physical barriers to interaction, potentially offering greater opportunities for engagement and preparation before medical procedures. Another recent study has highlighted the emerging role of metaverses in addressing psychiatric concerns in children and adolescents [29]. This study emphasized the potential of virtual environments to provide psychological support, which aligns with our approach. In our study, we specifically applied the concept of tailored metaverse-based interaction to pediatric and AYA patients with cancer in order to address the unique psychosocial needs of this population. These studies, along with our findings, underscore the potential of digital platforms to revolutionize pediatric healthcare delivery and support systems, particularly in creating accessible and engaging spaces for peer interaction and emotional support. Our study extends these findings by specifically focusing on the application of metaverse technology in oncology settings, demonstrating its potential to provide crucial support for a particularly vulnerable patient population.

### 4.6. Limitations

This study has two main limitations: its small sample size (n = 10) and the short-term nature of the intervention. These factors limit the generalizability of our findings and preclude any conclusions regarding long-term effects. Future research should address these limitations by including larger samples and implementing longer follow-up periods to assess the sustained impact of metaverse-based peer support interventions on pediatric and AYA cancer care.

## 5. Conclusions

This pioneering study is the first global initiative to investigate the impact of metaverse-mediated interactions on hospitalized cancer patients, underscoring the metaverse’s potential as an innovative healthcare solution [30,31,32]. The findings demonstrate that the metaverse can serve as a platform for continuous interaction and support, addressing the critical need for ongoing psychological support among cancer survivors post treatment [33,34,35]. Moreover, this study highlights the effectiveness of child life specialists [36,37], essential members of pediatric hospital care teams, in utilizing the metaverse to interact with patients across various facilities, thereby reducing patient stress and enhancing the quality of care. A metaverse-based support model improves treatment adherence and quality of life for patients with rare cancers and shows potential for broader applications across various medical fields. This marks a significant step towards enhancing healthcare quality and advancing patient-centered care. To implement the metaverse in hospitals or healthcare centers, the following steps are recommended. First, healthcare professionals should acquire a comprehensive understanding of the metaverse platform and provide thorough instructions to patients regarding its safe and effective use. In hospitals, the implementation process should include the establishment of dedicated VR rooms in which patients can participate privately. The necessary equipment, including VR goggles, tablets, and smartphones, should be provided. Furthermore, the medical team should organize regular metaverse sessions to promote interactions among patients. When introducing this tool to patients, healthcare professionals should highlight the psychological benefits and opportunities for peer connections offered by the metaverse. Specifically, they should elucidate how avatar-based interactions can mitigate anxiety related to physical changes resulting from treatment, and diminish feelings of isolation. Sharing success stories and patient feedback can further enhance trust in technology. Future research should focus on validating these findings on a larger scale and assessing the long-term psychological and social effects of ongoing metaverse interactions. Although this study was conducted in Japan, the inherent universality of the metaverse allowed us to overcome national and linguistic barriers. The integration of generative AI enables real-time translation and voice synthesis, facilitates seamless communication across various languages and cultures, and promotes international collaboration and multilingual engagement. This positions the metaverse as a truly global platform with the potential to revolutionize healthcare delivery worldwide. By enhancing mental health and facilitating adequate peer support, the metaverse can have a profound impact on pediatric and AYA oncology patients worldwide. The insights from this study advocate for the widespread adoption and implementation of metaverse initiatives in healthcare, leveraging their unique capabilities to transform patient care on a global scale.

## Figures and Tables

**Figure 1 cancers-16-02617-f001:**
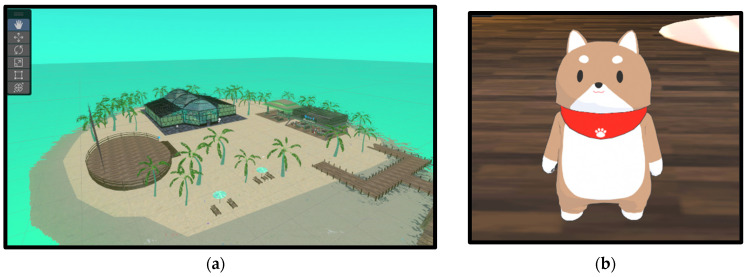
Examples of the design of the metaverse space and avatars for inpatient interaction: (**a**) This space was designed to reduce stress in children who have not been out of the hospital for extended periods. It incorporates elements of the natural environment with places such as the ocean, sandy beaches, corals, boats, and botanical gardens. (**b**) Dog avatars were used to reduce the presence of medical personnel.

**Figure 2 cancers-16-02617-f002:**
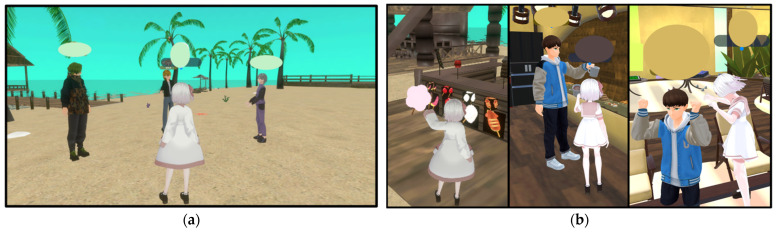
Scenes of interaction in the metaverse: (**a**) Conversations in a group. (**b**) Interface with various gimmicks—images of people holding cotton candy, toasting with mugs, and enjoying dancing.

**Table 1 cancers-16-02617-t001:** Patient characteristics.

Case	Age	Diagnosis	Treatment Status	City (Region)	Primary Site
1	10	Leukemia	Undergoing	Nagoya (Central)	-
2	15	Leukemia	Undergoing	Nagoya	-
3	19	Ewing sarcoma	Undergoing	Okayama (West)	Spine
4	14	Leukemia	Undergoing	Nagoya	-
5	20	Ewing sarcoma	Undergoing	Gifu (Central)	Spine
6	18	Ewing sarcoma	Just completed	Nagoya	Chest wall
7	15	Leukemia	Undergoing	Nagoya	-
8	22	Osteosarcoma	Survivor	Okayama	Femur
9	21	Osteosarcoma	Survivor	Yokohama (East)	Pelvis
10	15	Leukemia	Undergoing	Nagoya	-

**Table 2 cancers-16-02617-t002:** Feedback from Ewing’s sarcoma patients.

	Case 3	Case 5	Case 6
Is it easier to talk to someone using an avatar than to meet and talk to someone for the first time?	4	5	5
Is it easier to talk about your feelings if you speak as an avatar?	5	5	5
Compared to talking to actual friends, family members, etc., did you find the patient-to-patient metaverse exchanges to be a meaningful time?	5	5	5
There are patients who have experienced treatment for the same disease in the past and are now returning to social life. Would you like to hear their stories about their experiences during the time they were undergoing treatment and the difficulties and innovations they experienced in their social lives?	5	5	5
The people you have talked to and befriended in these metaverse exchanges are friends who fought the disease at the same time. Do you feel that you would like to meet them in person when your treatment is over?	5	5	5
You have spoken with healthcare professionals you have never met before and who were from a different facility—did you feel more comfortable speaking with them if they were an avatar?	5	5	5
Did you feel that talking with healthcare professionals had a positive effect on you, such as reducing stress or making you feel more comfortable?	5	5	5

## Data Availability

The data obtained in this study are not freely available to the public in order to protect the participants’ privacy. The data used in this study are available upon request from the corresponding author.

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
