# Peer review of "Utilizing the Metaverse to Provide Innovative Psychosocial Support for Pediatric, Adolescent, and Young Adult Patients with Rare Cancer"

_cancers, 2024, doi:10.3390/cancers16152617_

Round 1

Reviewer 1 Report

Comments and Suggestions for Authors

The manuscript is well written for the most part and, the article is very interesting.  This is to be appreciated. However, there are some significant changes that sound be made in order to improve its quality and scientific soundness. Below, are reported the points that can be improved:  

Materials and methods - Participants:

The way patients were recruited is unclear. It should be described where patients were recruited, age, location, and inclusion and exclusion criteria. Please better explain how patients were enrolled.

Discussion: The discussion and conclusions section need to be implemented with further studies.

You could cite the following articles:

1)     Ranieri J, Guerra F, Cilli E, Di Giacomo D. An Integrated Approach for a New Pattern in Pediatric Primary Care: Interaction Mediation for Active and Efficient Medical Consultations. Front Pediatr. 2020 Sep 4;8:530. doi: 10.3389/fped.2020.00530

2)     Kim S, Kim E. Emergence of the Metaverse and Psychiatric Concerns in Children and Adolescents. Soa Chongsonyon Chongsin Uihak. 2023 Oct 1;34(4):215-221. doi: 10.5765/jkacap.230047.

Author Response

Comment: The way patients were recruited is unclear. It should be described where patients were recruited, age, location, and inclusion and exclusion criteria. Please better explain how patients were enrolled.

Response: We appreciate your insightful comment regarding the need for clearer recruitment details. Your feedback has helped us improve the clarity and comprehensiveness of our methodology. In response to your suggestion, we have substantially revised the “Materials and Methods” section, specifically Subsection 2.1, as follows.

“The study involved ten cancer patients and survivors, aged 10-22 years, from four distinct oncology centers across Japan, representing both urban and rural areas, including isolated areas. This approach ensured comprehensive demographic representation and captured a broad spectrum of patient experiences. The inclusion criteria were as follows: (1) confirmed diagnosis of cancer, and (2) willingness to participate in virtual interactions. Exclusion criteria included (1) acute medical conditions precluding participation and (2) lack of parental consent for minors. Primary oncologists at each center identified potential participants based on these criteria from among pediatric, adolescent, and young adult patients with cancer. The research team subsequently approached eligible patients and their families for enrollment.”

Comment: The discussion and conclusions section need to be implemented with further studies. You could cite the following articles:

  • Ranieri J et al. An Integrated Approach for a New Pattern in Pediatric Primary Care: Interaction Mediation for Active and Efficient Medical Consultations. Front Pediatr. 2020; 8:530.
  • Kim S, Kim E. Emergence of the Metaverse and Psychiatric Concerns in Children and Adolescents. Soa Chongsonyon Chongsin Uihak. 2023; 34(4):215-221.

Response: Thank you for this valuable suggestion. We have expanded our discussion to include these relevant studies. The following paragraph has been added to the Discussion section:

4.5. Alignment with Recent Innovative Approaches in Pediatric Care

Our findings align with recent research on innovative approaches to pediatric care. Reports have demonstrated the efficacy of interactional mediation in enhancing medical consultations for children[28]. This study showed that children who engaged in interactive activities before medical consultations exhibited better cooperation during clinical care. This highlights the efficacy of the metaverse as a mediating platform for peer support, as both approaches aim to facilitate effective communication and cooperation in pediatric healthcare settings. The Metaverse extends this concept by providing a virtual environment that can overcome physical barriers to interaction, potentially offering greater opportunities for engagement and preparation before medical procedures. Recent studies have highlighted the emerging role of metaverses in addressing psychiatric concerns in children and adolescents[29]. This study emphasized the potential of virtual environments to provide psychological support that aligns with our approach. In our study, we specifically applied the concept of tailored metaverse-based interaction to pediatric and AYA patients with cancer to address the unique psychosocial needs of this population. These studies, along with our findings, underscore the potential of digital platforms to revolutionize pediatric healthcare delivery and support systems, particularly in creating accessible and engaging spaces for peer interaction and emotional support. Our study extends these findings by specifically focusing on the application of metaverse technology in oncology settings, demonstrating its potential to provide crucial support for a particularly vulnerable patient population.”

We believe that this addition strengthens our discussion by placing our findings in the context of current research trends and highlighting the innovative aspects of our study.

Reviewer 2 Report

Comments and Suggestions for Authors

I want to thank you for the opportunity to review this study.

I believe it is very important to investigate the psychological well-being that social support, especially peer support, provides in cases of recovery from cancers or other illnesses. There is scientific literature on this topic, but including the metaverse is a novel and intriguing aspect for children and adolescents. I think this is a great move.

The introduction adequately captures this idea and is well-referenced. Regarding the methods section, specifically in the participants subsection, it would be beneficial to provide more detailed descriptions of the participants' characteristics, such as their age and type of illness. This information appears in the results section, but it pertains to participant characteristics rather than study outcomes.

In the procedure section, it would be necessary to specify how contact with the families was established, possibly through hospitals, oncologists, or other means.

In the intervention section, it is essential to specify whether the metaverse space was specifically designed for this study or if an existing space was chosen. If it was designed for this study, highlighting the team involved in its design (computer scientists, psychologists, etc.) would be interesting. Additionally, describing how the intervention was conducted, such as using tablets, computers, mobile devices, and whether it took place in patients' homes, hospitals, or elsewhere, is crucial.

The results and discussion are explained very clearly. It might be beneficial to include a paragraph discussing the study's limitations.

Finally, in the conclusion, it would be valuable to include how this tool could be applied in hospitals or centers, or how healthcare professionals could introduce it to people suffering from these illnesses.

Overall, it's a very interesting study.

Author Response

Comment: Regarding the methods section, specifically in the participants subsection, it would be beneficial to provide more detailed descriptions of the participants' characteristics, such as their age and type of illness.

Response: We appreciate the reviewer's suggestion regarding participant characteristics. We agree that providing detailed information about participants is crucial. To address this issue, we made the following changes:

In the Methods section, we have added more detailed information about the recruitment process and the inclusion/exclusion criteria. This included the age range of potential participants (10-22 years), geographical distribution of the oncology centers, and specific criteria used for participant selection.

The detailed characteristics of the study participants, including their specific ages and types of cancer, are presented in the Results section. We believe this is the most appropriate location for this information as it represents the outcome of our recruitment process rather than the methodology itself.

To clarify the distinction between the recruitment criteria and actual participant characteristics, we have added the following transitional sentence at the beginning of the Results section: “Based on the recruitment criteria outlined in the Methods section, the following participants were enrolled in the study.”

We believe that this approach maintains the logical flow of the paper while providing all the necessary information about both the intended participant pool and the actual study cohort. This structure allows readers to understand our methodological approach to participant selection and resulting study populations.

Comment: In the procedure section, it would be necessary to specify how contact with the families was established, possibly through hospitals, oncologists, or other means.

Response: Thank you for highlighting this important aspect of the methodology. We appreciate your attention to detail, which has helped us provide a clearer picture of our recruitment process. We have addressed this by modifying the procedure section as follows:

We have added information regarding the initial contact process at the beginning of Subsection 2.2. Procedure (Setting and Recruitment). The revised section is as follows.

“Initial contact with potential participants and their families was established by the treating oncologists at each participating hospital. The research team then conducted sessions at hospitals to explain the study's objectives and procedures to interested families. These initial briefings were held to inform patients and their guardians about the objectives and framework of the metaverse interaction sessions.”

This addition provides a clear explanation of how families were initially approached and how they were informed about the study, addressing the reviewers’ concerns while maintaining the flow of information in the existing text.

Comment: In the intervention section, it is essential to specify whether the metaverse space was specifically designed for this study or if an existing space was chosen. If it was designed for this study, highlighting the team involved in its design (computer scientists, psychologists, etc.) would be interesting.

Response: Thank you for your insightful comment on the design of the metaverse space. Your suggestion allowed us to provide more comprehensive information regarding our interventions. We have expanded subsection 2.3. Intervention to address this point: “The metaverse space used in this study was specifically designed for this research through a collaborative effort of medical professionals and metaverse designers. The virtual environment was developed using Unity as a world within "cluster", a metaverse platform provided by Cluster, Inc. This platform was chosen for its accessibility across various devices, including smartphones, PCs, and VR headsets, ensuring that all participants could engage regardless of their physical limitations or preferred mode of access. The initial 3D model of the space was created by professional metaverse designers. This basic framework was refined and customized by a multidisciplinary medical team consisting of specialists in bone and soft tissue tumors, pediatric oncology, and palliative care. These medical experts were responsible for the detailed arrangement of elements within the space and the creation of specific content tailored to the needs of pediatric and AYA cancer patients. The resulting virtual environment was enriched with natural elements, aiming to mitigate the lack of natural exposure for patients predominantly confined to hospital settings by offering a virtual yet immersive natural experience (Figure 1a). The platform's features, including real-time voice chat, emotes, and gesture animations, were utilized to enhance communication and interaction among participants.”

We believe that this addition provides valuable insights into the collaborative and tailored nature of our intervention design, as well as the technical aspects of the platform used.

For the Acknowledgments section, I suggest adding the following:

Acknowledgments: We also thank Cluster, Inc. for providing the "cluster" metaverse platform used in this study. Their technology was instrumental in creating an accessible and engaging virtual environment for our research participants.

Comment: Additionally, describing how the intervention was conducted, such as using tablets, computers, mobile devices, and whether it took place in patients' homes, hospitals, or elsewhere, is crucial.

Response: Thank you for your important observation. We appreciate your consideration of the practical aspects of our intervention, which are crucial for a comprehensive understanding of our study. We have added the following details to Subsection 2.3: Intervention:

“The method of accessing the metaverse varied depending on the participants' circumstances. For inpatients, those in private rooms participated from their hospital rooms, whereas those in shared rooms were moved to separate private spaces for the sessions. All inpatients used VR goggles provided by the research team to access the metaverse device. These devices are equipped with the necessary software and configured to ensure ease of use and privacy. Survivors who were no longer hospitalized participated from their homes using their own smartphones.”

This addition provides a clear picture of how and where the intervention was conducted, addressing the diverse circumstances of the participants.

Comment: It might be beneficial to include a paragraph discussing the study's limitations.

Response: We appreciate your suggestion to discuss the limitations of the study. This is a crucial aspect of any research, and we thank you for bringing it to our attention. We have addressed this by adding a new subsection (Section 4.6. Limitations) at the end of the discussion section. The added content is as follows:

“This study has two main limitations: a small sample size (n=10) and the short-term nature of the intervention. These factors limit the generalizability of our findings and preclude conclusions regarding long-term effects. Future research should address these limitations by including larger samples and implementing longer follow-up periods to assess the sustained impact of metaverse-based peer support interventions on pediatric and AYA cancer care.”

We believe that this addition provides a balanced view of our study and sets the foundation for future research in this area.

Comment: In the conclusion, it would be valuable to include how this tool could be applied in hospitals or centers, or how healthcare professionals could introduce it to people suffering from these illnesses.

Response: We are grateful for your insightful suggestion to include practical implementation details in the Conclusion section. This addition significantly enhanced the applicability of our research. We have expanded our conclusion to address this point: “To implement metaverse in hospitals or healthcare centers, the following steps are recommended. First, health care professionals should acquire a comprehensive understanding of the metaverse platform and provide thorough instructions to patients regarding its safe and effective use. In hospitals, the implementation process should include the establishment of dedicated VR rooms in which patients can participate privately. The necessary equipment, including VR goggles, tablets, and smartphones, should be provided. Furthermore, the medical team should organize regular metaverse sessions to promote interactions among patients. When introducing this tool to patients, healthcare professionals should highlight the psychological benefits and opportunities for peer connections offered by the metaverse. Specifically, they should elucidate how avatar-based interactions can mitigate anxiety related to physical changes resulting from treatment, and diminish feelings of isolation. Sharing success stories and patient feedback can further enhance trust in technology.”

This addition provides practical guidance for health care professionals seeking to implement metaverse-based interventions in their practice, thereby increasing the potential impact of our research.

Round 2

Reviewer 2 Report

Comments and Suggestions for Authors

Thank you very much for having taken into account all the suggestions made to your study in my review. And thank you also for the letter with your answers. I think the manuscript is ready to be published in the journal. It is a great work. I wish you all the best.